# A State-Based Peridynamic Flexural Fatigue Model for Contact and Bending Conditions

**DOI:** 10.3390/ma15217762

**Published:** 2022-11-03

**Authors:** Junzhao Han, Hao Yu, Jun Pan, Rong Chen, Wenhua Chen

**Affiliations:** 1School of Mechanical Engineering, Zhejiang Sci-Tech University, Hangzhou 310018, China; 2CSSC Systems Engineering Research Institute, Beijing 100094, China

**Keywords:** peridynamic fatigue model, flexural fatigue crack, crack initiation and propagation, fatigue life prediction

## Abstract

To address flexural fractures and predict fatigue life, an ordinary state-based peridynamic (PD) fatigue model is proposed for the initiation and propagation of flexural fractures. The key to this model is to replace the traditional partial differential fracture model with a spatially integral peridynamic model. Based on the contact and slip theory, the nonlocal peridynamic contact algorithm is confirmed and the load transfer is through the contact area. With the 3D peridynamic *J*-integration and the energy-based bond failure criterion, the peridynamic fatigue model for flexural cracks’ initiation and propagation is constructed. The peridynamic solid consists of a pair of gear contact surfaces and the formation and growth of flexural fatigue cracks evolved naturally over many loading cycles. The repeated load is transferred from the drive gear to the follower gear using the nonlocal peridynamic contact algorithm. The improved adaptive dynamic relaxation approach is used to determine the static solution for each load cycle. The fatigue bending crack angle errors are within 2.92% and the cycle number errors are within 10%. According to the experimental results, the proposed peridynamic fatigue model accurately predicts the location of the crack without the need for additional criteria and the fatigue life predicted by the simulation agrees quite well with the experimental results.

## 1. Introduction

Contact surfaces are common in both natural and constructed systems and they are frequently linked to structural and material failures, including pitting and flexural cracking [1,2]. Therefore, material fractures caused by periodic contact stresses need to be investigated both in terms of flexural crack growth and prediction of fracture behavior. In a gear transmission, contact between gears occurs when a pair of gears transmits power in different operating environments. Under the continuous action of the internal mechanism and the external environment, the Hertzian alternating stress on the tooth surface causes periodic bending stresses at the tooth root, which leads to the development of fatigue cracks at the tooth root surface [3,4]. The entire process of tooth fracture formation, including crack initiation, propagation, and final fracture, must be considered. In recent years, contact stress has increased, leading to more fatigue cracks on the tooth surface and root, which results in considerably greater maintenance costs. The gear may eventually fracture and even break if prompt action is not taken to stop the crack from spreading, seriously endangering its effectiveness and safety [5,6]. Due to the uncertainty of loads and material properties, predicting the initiation and propagation of fatigue in a structure is a difficult task. The fatigue damage law should be derived to ensure safe operation and reduce costs [7,8]. To ensure the effectiveness of the gear and lower maintenance costs, it is crucial to investigate the characteristics of fracture propagation at the tooth root.

Many scholars around the world have studied this typical fatigue damage of a gear under cyclic loading, mainly focusing on the crack initiation, crack propagation and crack development patterns of tooth fracture with a series of experimental studies [9,10]. Domestic and foreign scholars have carried out experimental studies on the fatigue of gears through chemical analysis, metallographic investigation, hardness measurement and other methods. The basic idea is to divide the fatigue life of gears into the life at crack initiation and the life at crack propagation and calculate both. The final fatigue life of a gear is obtained by adding the two values. The contact fatigue life of gears is mainly calculated by an analytical method and the flexural fatigue life of the tooth root is calculated by a combination of analytical method and the finite element method [11,12,13].

Thanks to advances in computer technology, numerical simulation techniques are now frequently used to reproduce and explain phenomena observed in experiments [14,15]. Numerous experimental studies are increasingly using numerical models based on classical continuum mechanics, such as the finite element approach [16] and boundary element method [17]. This method overcomes the shortcomings of conventional gear life calculations, such as insufficient data on the fatigue properties of gear materials and inaccurate stress calculations. The continuous 3D body is first discretized into a finite number of small elements. All displacements and forces are calculated using nodes, where the nodes connect the individual elements [18,19]. In each discretized element, a suitable interpolation function must be chosen at the inner boundary (subdomain interface) and the outer boundary (subdomain and outer interface) of the subdomain that satisfies certain conditions in that domain.

Currently, computer techniques for simulating the continuity discontinuity problem rely on the standard continuum mechanics partial differential equations. However, for discontinuities, such as crack branching in solid materials and structures, conventional numerical methods have the problem of singularity and low computational power [20,21]. The use of lattice reconstruction or the method of inserting a cohesive element into the finite element would lead to lattice-dependent results [22]. The partition algorithm and the notional crack model used by the boundary element method (BEM) have similar limitations to the finite element method in the analysis of crack propagation problems [23]. As a result, academics have suggested the extended finite element approach. It lessens the tight constraints on the mesh discontinuity when compared to the conventional finite element approach [24,25]. However, in constructing the extension function, the extended finite element method needs to know in advance the characteristics of the problem to be solved. This condition is relatively difficult for complex problems, such as crack branching and multiple-crack intersections.

Due to the spatial partial differential equations used to model the motion of the particle under consideration, the stress at the fracture tip is mathematically single [26,27]. As a result, crack initiation and propagation are individually modelled using different criteria. Determining the correlations between the damage parameters at the fatigue crack tip that describe the damage progression under cyclic loading is the main goal of the extended finite element approach and numerous other modified damage models [28,29]. The fundamental issue is that fatigue crack tips or crack faces cannot be directly applied to these models. The body in each calculation method was redefined and extra equations were included so that the additional assessment criteria for evaluating fracture propagation and crack angle are obtained [30,31]. Corrective actions were taken for the current techniques.

The use of the stress separation rule for the opening model I, the fracture mechanism and the mixing model under external loading to solve these problems is a milestone in computational mechanics [32,33]. The interfaces in the material are modeled using the provided law and cohesion zone elements are placed on the ill-defined region using conventional methods. The development of fatigue cracks demonstrates the property of mesh dependence, which requires the use of a special dehumidification technique. However, the mesh structure and density affect the rate of crack initiation. Even in simple cracking models, the solid-state dehumidification process is often complex, making convergence of results difficult [34,35]. The extended finite element approach is proposed as a compromise to eliminate meshing in fatigue crack propagation to circumvent these difficulties. Its typical features include the probability of the crack propagating across the element surface and a local strengthening effect. The need to construct additional control equations for factors, such as fatigue crack location, fracture angle and direction, crack extension and arrest under external loading, does not affect the model FEM, various modified versions or the XFEM method [36,37]. In other words, the standard framework of continuum theory is used to treat the onset and extension of fatigue cracks with a one-sided treatment strategy that incorporates various mathematical systems. Dislocations and grain boundaries directly govern the entire process of fatigue fracture and this is where most microcracks occur [38,39]. They range in size and duration from microscopic to macroscopically obvious cracks. Therefore, it is challenging to predict the location of fatigue nucleation using a numerical solution and to draw this conclusion from a complicated and unpredictable testing technique [40,41,42].

Although many laws and equations have been derived to describe the two phases of the evolution of cracks and fissures, the key to the cracking problem in the field of classical local continuum theory is still unsolved and complicated. The crucial and contradictory step in this framework is the application of the continuum equation to discontinuities in the body, which leads to the contradictory results [43,44]. The material structure control equation cannot handle processes that lead to fractures or other discontinuities and numerous solutions are proposed to remove the discontinuity from the original framework. This process promotes fuzziness and ambiguity, especially when cracks develop and propagate [45]. The size of the element and the way the edges are treated have a significant impact on traditional numerical techniques, such as FEM and XFEM. A mesh-free method based on continuum mechanics was developed to reduce this dependence and obtain accurate results. Two rules [46] are satisfied by the new mesh damage model. First, at the crack tip without unique stress and strain values, where the transition from the continuous phase to the discontinuous phase is smooth, accurate numerical solutions are normal. Secondly, there is excellent agreement between the fracture model and crack progression and empirical data collected at the research facility.

To resolve the contradiction between the continuity assumption and the discontinuity phenomenon of the failure problem, Silling [47] proposed a nonlocal method called peridynamics (PD) to describe the motion process of material particles. In contrast, a peridynamic theory (PD) calculates the internal force acting on a material particle using spatial integral equations rather than derivatives of the displacement field [48,49,50]. One component in the constitutive model is material damage. Peridynamics allows for crack propagation at numerous locations with natural paths, not just along the element boundary in a coherent framework, without requiring specific criteria for crack growth. To represent the progression of damage in a peridynamic solid, this nonlocal theory reconstructs the particle control equations using a novel model. The integral representation solves the discontinuity problem more effectively than partial differential forms when the given particle is in contact with other neighbors in a finite region, the peridynamic radius. Peridynamics has the advantage of predicting the type of cracks that can occur in fatigue damage under cyclic loading. Material damage begins and propagates naturally; therefore, additional criteria are unnecessary. Moreover, due to the novel particle interactions between the peridynamic solid, the path of the crack is arbitrary, unlike the classical frame that propagates only along the finite element boundary. The PD theory can connect the micro-length scale with the macro-length scale [51,52,53]. The results show that peridynamics has no singularity problem in the analysis of the failure problem and can simulate the whole process of the material, including macroscopic crack initiation, propagation and final failure. The above simulation of failure is based on bond-based peridynamics theory (BBPD). However, the bond-based peridynamic theory has some problems, such as Poisson’s ratio restriction and lack of connection with traditional continuum theory. Silling et al. [54] proposed an ordinary state-based peridynamic theory (OSBPD) and a non-ordinary state-based peridynamic theory (NOSBPD). Both of them adopt the advantages of BBPD for solving discontinuous problems and have a similar definition of state variables as traditional physical variables. In recent years, many researchers have started to study the contact model based on the PD method. Madenci et al. [55] developed a NOSBPD model for brittle fracture to simulate edge impact and drop ball test and discussed the contact algorithm between projectile and target. Littlewood et al. [56] summarized the simulation results of the finite element method and peridynamics. A combined approach of finite element method and peridynamics via a contact algorithm is used. Kamensky et al. [57] summarized several existing peridynamic contact friction models and introduced a state-based nonlocal friction formulation to demonstrate the properties of different peridynamic contact models using some impact and penetration problems. In the case of small deformations, this model agrees with the classical Hertzian contact analysis. Silling et al. [58] proposed a new PD model to simulate elastoplastic behavior, creep and fracture.

Lengths in different sizes, from micro to macro, are included in the damage model PD during fatigue loading. Without making any special assumptions, we can use this formulation to model fracture initiation, propagation, branching and coalescence. Oterkus and Madenci [59] first introduced the peridynamic fatigue damage law, while Nguyen et al. [60] then proposed a modified version based on a fundamental physical theory. A continuous model was used to represent the whole process of fatigue damage due to cyclic loading and the results of conventional fatigue tests were used to validate the characteristic fatigue parameters. The damage model in [61,62] deals with the onset and propagation of fatigue cracks under cyclic loading. A suitable damage law is derived from the S–N curve and Paris law, which causes crack initiation and propagation. The test results show that the peridynamic simulation results agree very well with the decrease in stiffness and strength. The results show that the usual fatigue damage peridynamic model is capable of handling the specifics of fracture initiation and propagation without the need for additional rules [63,64].

In this study, a nonlocal peridynamic contact technique and 3D J peridynamic integration are introduced to propose a new bending-fatigue damage model based on OSPD theory for a gear pair. The arrangement of the remaining sections is given below. The nonlocal peridynamic contact algorithm between two contact surfaces based on OSPD theory and contact and slip theory is briefly described in Section 2. In Section 3, the peridynamic fatigue model for gear pairs is established and the bending stress at the tooth root is determined by the interfacial loading transformation. The peridynamic criteria for crack initiation and propagation are established. The failure process of a tooth fracture under periodic bending loading is simulated in Section 4 along with an explanation of the overall numerical calculation methodology. The static solution for each loading cycle is derived using the improved adaptive dynamic relaxation approach. The convergence analysis for two pairs of gears operating under different loading conditions is described in Section 5. The experimental results show that the proposed method effectively captures the crack-sensitive region without the need for additional criteria. The fatigue life obtained in the simulation agrees quite well with the experimental results, which proves the effectiveness of the proposed approach.

## 2. Nonlocal State-Based Peridynamic Contact Algorithm

In this part, we construct a new nonlocal version of the state-based peridynamic contact model, describe peridynamic contact connections for nonlocal contact modeling and develop contact forces to treat rolling and sliding contact conditions.

### 2.1. State-Based Peridynamic Theory

State-based peridynamic theory uses displacements instead of displacement derivatives in its spatial governing equations. A material particle enters into connections with other particles via the prescribed reaction function in a nonlocal region. Based on the principle of virtual work, the equation for the motion control of a material particle $x_{\left(k\right)}$ can be expressed as follows:(1)$$\left\{ \begin{array}{l} \frac{d}{dt}[\frac{\partial L}{\partial {\dot{u}}_{\left(k\right)}}]-\frac{\partial L}{\partial u_{\left(k\right)}}=0\\ L=T-U \end{array}\right.$$
where L is the Lagrangian function and T and U denote the total kinetic and potential energy of the body. If you add the kinetic and potential energy of all material particles, you can calculate the total kinetic and potential energy of the body.
(2)$$\left\{ \begin{array}{l} T=\sum_{i=1}^{\infty }\frac{1}{2}\rho_{\left(i\right)}{\dot{u}}_{\left(i\right)}\cdot {\dot{u}}_{\left(i\right)}V_{\left(i\right)}\\ U=\sum_{i=1}^{\infty }W_{\left(i\right)}V_{\left(i\right)}-\sum_{i=1}^{\infty }\left(b_{\left(i\right)}\cdot u_{\left(i\right)}\right)V_{\left(i\right)} \end{array}\right.$$

The following expression can be used to replace the strain energy density $W_{\left(i\right)}$ of the material point $x_{\left(i\right)}$.
(3)$$\begin{array}{c} W_{\left(k\right)}=\frac{1}{2}\overset{\infty }{\underset{j=1}{\sum }}\frac{1}{2}(w_{\left(k\right)\left(j\right)}\left(y_{\left(1^{k}\right)}-y_{\left(k\right)},y_{\left(2^{k}\right)}-y_{\left(k\right)},\cdots \right)\\ +w_{\left(j\right)\left(k\right)}\left(y_{\left(1^{j}\right)}-y_{\left(j\right)},y_{\left(2^{j}\right)}-y_{\left(j\right)},\cdots \right))V_{\left(j\right)} \end{array}$$
where $w_{\left(k)(j\right)}=0\;for\;k=j$, then the potential energy can be expressed as:(4)$$U=\sum_{i=1}^{\infty }\left\{\frac{1}{2}\sum_{j=1}^{\infty }\frac{1}{2}\left[\begin{array}{c} w_{\left(i)(j\right)}\left(y_{\left(1^{i}\right)}-y_{\left(i\right)},y_{\left(2^{i}\right)}-y_{\left(i\right)},\cdots \right)\\ +w_{\left(j)(i\right)}\left(y_{\left(1^{j}\right)}-y_{\left(j\right)},y_{\left(2^{j}\right)}-y_{\left(j\right)},\cdots \right) \end{array}\right]V_{\left(j\right)}-\left(b_{\left(i\right)}\cdot u_{\left(i\right)}\right)\right\}V_{\left(i\right)}$$

Using Equation (1), the Lagrangian can be stated in expanded form by defining only the terms related to the material point, $x_{\left(k\right)}$.
(5)$$\begin{array}{l} L=\ldots +\frac{1}{2}\rho_{\left(k\right)}{\dot{u}}_{\left(k\right)}\cdot {\dot{u}}_{\left(k\right)}V_{\left(k\right)}+\cdots \\ \cdots -\frac{1}{2}\sum_{j=1}^{\infty }\{\frac{1}{2}[w_{\left(k)(j\right)}\left(y_{\left(1^{k}\right)}-y_{\left(k\right)},y_{\left(2^{k}\right)}-y_{\left(k\right)},\cdots \right)\\ +w_{\left(j)(k\right)}\left(y_{\left(1^{j}\right)}-y_{\left(j\right)},y_{\left(2^{j}\right)}-y_{\left(j\right)},\cdots \right)]V_{\left(j\right)}\}V_{\left(k\right)}\cdots \\ \cdots -\frac{1}{2}\sum_{i=1}^{\infty }\{\frac{1}{2}[w_{\left(i)(k\right)}\left(y_{\left(1^{i}\right)}-y_{\left(i\right)},y_{\left(2^{i}\right)}-y_{\left(i\right)},\cdots \right)\\ +w_{\left(k)(i\right)}\left(y_{\left(1^{k}\right)}-y_{\left(k\right)},y_{\left(2^{k}\right)}-y_{\left(k\right)},\cdots \right)]V_{\left(i\right)}\}V_{\left(k\right)}\cdots \\ \ldots +\left(b_{\left(k\right)}\cdot u_{\left(k\right)}\right)V_{\left(k\right)}\cdots \end{array}$$

As can be seen in Figure 1a, the material point $x_{\left(k\right)}$ interacts directly and nonlocally with all points that lie within a distance δ of $x_{\left(k\right)}$. We call δ the horizon and refer to the spherical region with radius δ centered at $x_{\left(k\right)}$ as $H_{x_{\left(k\right)}}$, the family of $x_{\left(k\right)}$. As can be seen in Figure 1b, the deformation at $x_{\left(k\right)}$ depends collectively on the deformations of $H_{x_{\left(k\right)}}$. The motion equation of the material particle $x_{\left(k\right)}$ in the deformed configuration is revised. Equation (5) is substituted into Equation (1) to obtain the following Lagrangian equation for the material point $x_{\left(k\right)}$:(6)$$\begin{array}{c} \rho (x_{\left(k\right)})\ddot{u}(x_{\left(k\right)},t)=\int_{H_{x_{\left(k\right)}}}\{\underline{T}[x_{\left(k\right)},t]\langle x_{\left(j\right)}-x_{\left(k\right)}\rangle -\underline{T}[x_{\left(j\right)},t]\langle x_{\left(k\right)}-x_{\left(j\right)}\rangle \}d{ℋ}_{\left(x_{\left(k\right)}\right)}\\ \;\;\;\;\;\;+b(x_{\left(k\right)},t) \end{array}$$
where ρ represents the mass density, $\underline{T}$ is the force state described below, the angle brackets to indicate the vector $x_{\left(j\right)}-x_{\left(k\right)}$ on which the state $\underline{T}$ acts and b is an external body force density. The corresponding peristatic equation can be expressed as follows:(7)$$-\int_{H_{x_{\left(k\right)}}}\left(\underline{T}[x_{\left(k\right)},t]\langle x_{\left(j\right)}-x_{\left(k\right)}\rangle -\underline{T}[x_{\left(j\right)},t]\langle x_{\left(k\right)}-x_{\left(j\right)}\rangle \right)d{ℋ}_{\left(x_{\left(k\right)}\right)}=b(x_{\left(k\right)},t)$$

Each point $x_{\left(k\right)}$ of the peridynamic body interacts directly with each point of the radius sphere $H_{x_{\left(k\right)}}$, as shown in Figure 2a (the family of $x_{\left(k\right)}$). In its deformed state $\underline{Y}$, the deformation state eventually forms a bond ξ. The relative displacement between a material particle and the other material particles within its horizon determines the force state for that material point. Therefore, the force state can be expressed as follows:(8)$$\underline{T}[x_{\left(k\right)},t]=\underline{T}(\underline{Y}[x_{\left(k\right)},t])$$

All relative position vectors in the particle’s horizon $x_{\left(k\right)}$, $y_{\left(j\right)}-y_{\left(k\right)}\;(j=1,2,\cdots ,\infty )$, as depicted in Figure 2b, are described as follows:(9)$$\underline{Y}\left(x_{\left(k\right)},t\right)=\left\{\begin{array}{c} y_{\left(1\right)}-y_{\left(k\right)}\\ \vdots \\ y_{\left(\infty \right)}-y_{\left(k\right)} \end{array}\right\}$$
where $\underline{Y}\left(x_{\left(k\right)},t\right)$ is the deformation vector’s current state. According to Equation (5), a material point’s $x_{\left(k\right)}$ overall reaction is dependent on the deformation of all bonds that are connected to the particle.

### 2.2. Contact Criteria for Sliding and Rolling between Two Surfaces

As shown in Figure 3a, the contact between two physical surfaces is nonlinear and discontinuous. Based on Hertz’s contact theory, the contact area between the two surfaces is expressed as follows:(10)$$R_{a}=\sqrt{\frac{3P}{4}\frac{\frac{1-\mu_{1}^{2}}{E_{1}}+\frac{1-\mu_{2}^{2}}{E_{2}}}{\frac{1}{a_{1}}+\frac{1}{a_{2}}}}$$
where $a_{1}$ and $a_{2}$ are radius of curvature, $\mu_{1}$ and $\mu_{2}$ are the Poisson ratios, $E_{1}$ and $E_{2}$ are elastic modulus of elasticity and $R_{a}$ is the radius of the contact area.

As can be seen in Figure 3b, the peridynamic numerical methods artificially plot the interfaces of the gate and contactor surfaces by extending to $0.5d_{1}$ and $0.5d_{2}$ from the nodes of the interfaces along their outer ordinary unit vectors $n_{1}$ and $n_{2}$, where $d_{1}$ and $d_{2}$ are the lattice sizes of the peridynamic gate and contactor bodies. In peridynamic modes, since there must be no overlap of substances for contact to occur, contact occurs when the following conditions are satisfied:(11)$$\left\{ \begin{array}{l} d_{x}=(y_{i}-y_{k})\cdot n\leq \frac{1}{2}(d_{1}+d_{2})\\ \left|y_{i}-y_{k}\right|\leq \frac{\sqrt{2}}{2}(d_{1}+d_{2}) \end{array}\right.$$
where n is the unit normal vector of the contact surface, $y_{i}$ and $y_{k}$ are the deformed vectors of the boundary particles $x_{i}$ and $x_{k}$, $x_{i}$ and $x_{k}$ being the outermost nodes belonging to contactor body and target body, respectively, and $d_{x}$ is the distance between two dashed parallel lines.

If inequalities (11) are satisfied, the particle $x_{i}$ of the interface is in its contact region $H_{ic}$. As for the boundary node $x_{i}$ itself, $\delta_{c}$ is the radius of the peridynamic horizon for the calculation of the contact force and $x_{k}$ is the particle on the boundary target body. The contact region $H_{ic}$ is defined as follows:(12)$$\left\{ \begin{array}{l} h=a_{2}-\sqrt{{\left(a_{2}\right)}^{2}-{\left(\frac{1.5R_{a}}{2}\right)}^{2}}\\ S_{H_{ic}}=2\pi a_{2}h \end{array}\right.$$
where $R_{a}$ is calculated in Equation (6) and $S_{H_{ic}}$ is the surface area of the contact region $H_{ic}$. The particle $x_{k}$ that belongs to the virtual target surface in the region $H_{ic}$ and the contact horizon $\delta_{c}$ is defined. Once the surface of the contact region $H_{ic}$ is confirmed, the number of all boundary contact nodes of the virtual target surface can be confirmed.

### 2.3. State Forces in Contact Region

The peridynamic contact bond $\xi_{ik}^{c}$, which is illustrated in Figure 4a, describes the connection between the boundary contact node and its contact nodes and is defined as a thick dashed line to distinguish it from the true peridynamic bond. The peridynamic contact bond $\xi_{ik}^{c}$ can be defined as follows:(13)$$\xi_{ik}^{c}=x_{k}-x_{i}$$
where $\underline{x}\langle \xi_{ik}^{c}\rangle$ and $\underline{Y}\langle \xi_{ik}^{c}\rangle$ are reference vectors and deformed vectors in the contact region. The bond $\xi_{ik}^{c}$ deforms when contact occurs and two forms of deformed states are described as:(14)$$\left\{ \begin{array}{l} \underline{T_{t}}\langle \xi_{ik}^{c}\rangle =\frac{\left|\underline{Y}\langle \xi_{ik}^{c}\rangle \right|-\left|\underline{X}\langle \xi_{ik}^{c}\rangle \right|}{\left|\underline{X}\langle \xi_{ik}^{c}\rangle \right|}\\ \underline{T_{n}}\langle \xi_{ik}^{c}\rangle =\frac{\left|\underline{Y}\langle \xi_{ik}^{c}\rangle \cdot n\right|-\left|\underline{X}\langle \xi_{ik}^{c}\rangle \cdot n\right|}{\left|\underline{X}\langle \xi_{ik}^{c}\rangle \cdot n\right|} \end{array}\right.$$
where $\underline{T_{t}}\langle \xi_{ik}^{c}\rangle$ is the bond stretch along the deformed bond directions and $\underline{T_{n}}\langle \xi_{ik}^{c}\rangle$ is the bond normal stretch along the bond normal direction, which is also the tangential directions of the contact surface.

In this nonlocal contact calculation method, two forms of peridynamic contact bond forces are defined for two exclusive contact cases. As for sticking cases, the contact bond force is described as follows:(15)$$F_{\underline{T_{t}}}\langle \xi_{ik}^{c}\rangle =c_{T_{t}}\underline{T_{t}}\langle \xi_{ik}^{c}\rangle \omega \langle \xi_{ik}^{c}\rangle \frac{\underline{Y}\langle \xi_{ik}^{c}\rangle }{\left|\underline{Y}\langle \xi_{ik}^{c}\rangle \right|}$$

As shown in Figure 4b, sliding friction forces are divided into peridynamic regular and tangential adhesion forces. These two types of adhesion forces can be expressed as follows:(16)$$\left\{ \begin{array}{l} \underline{F_{n}}\langle \xi_{ik}^{c}\rangle =-c_{T_{n}}\underline{T_{n}}\langle \xi_{ik}^{c}\rangle \underline{\omega }\langle \xi_{ik}^{c}\rangle n\\ \underline{F_{f}}\langle \xi_{ik}^{c}\rangle =\mu \left|\underline{F_{n}}\langle \xi_{ik}^{c}\rangle \right|e \end{array}\right.$$
where $\underline{T_{t}}$ and $\underline{T_{n}}$ are the deformed bond and the bond normal strain of the bond from Equation (14), μ is the function coefficient, $\underline{\omega }$ is the influence function, n and e are the unit normal and tangent vectors of the contact floor, respectively, and e is equal to the slip distance. For the peridynamic contact bond, the sticking and sliding micromodulus are $c_{T_{t}}$ and $c_{T_{n}}$, respectively. The detailed derivation of the two parameters is shown in the Appendix A.

In general, the manifestations of the peridynamic bond force are exceptional for the sticking and sliding contact. The sticking contact bond can be considered as a true peridynamic compressive force, because the strain $\underline{T_{t}}$ is used to calculate the bond pressure computation. Along the deformed vector, the force is applied. Just like the sliding friction contact, the tangential component of the bond deformation does not contribute to the bond forces and does not affect the contact bond forces when calculating the normal strain $\underline{T_{n}}$. In addition, the contact bond forces for each sticking and sliding case are limited by Newton’s Third rule.
(17)$$\underline{F}\langle \xi_{ik}^{c}\rangle =-\underline{F}\langle \xi_{ki}^{c}\rangle$$
where this guarantees the requirement of linear admissibility and provides the bond force structures of the contact area nodes.

### 2.4. Contact Algorithm between Two Discrete Peridynamic Bodies

The groups of nodes near the boundary are confirmed for the practice of contact evaluation practice. If the contact condition is satisfied during each iteration step, the sticking force and the sliding normal force are calculated and the normal vectors of the contact region are then checked. The sticking contact forces for static and dynamic frictional contact are calculated. The whole process of the nonlocal state-based peridynamic contact algorithm can be illustrated, as shown in Figure 5.

## 3. Peridynamic Fatigue Model

In this phase, a model for peridynamic fatigue is developed that is state based and focuses primarily on the mechanism of fatigue. When a joint first fails during the cracking phase, subsequent joints experience progressive deterioration due to repeated bending loads greater than the local fatigue strength in the peridynamic solid. Under different fatigue loads, each bond in the peridynamic solid is characterized as an ideal fatigue specimen. In a peridynamic solid, the bond points are connected by physical interactions. The physical interactions are permanently extinguished when the bonds break within the confined region, the irrevocable breaking of the spring-like bond between the two particles, m and j. As a result, the fatigue load is redistributed within the peridynamic solid at each loading cycle, causing progressive fatigue damage that propagates independently.

### 3.1. Damage Models for Peridynamic Bond and Material Particle

The particles $x_{j}$ and $x_{m}$ sustain progressive damage as a result of the failure of bond $\xi_{jm}$ in the peridynamic solid body, as shown in Figure 6a.
(18)$$d_{jm}\left(\xi ,t\right)=\{\begin{array}{c} 1\;if\;\;\xi_{jm}\;\;is\;\;broken\\ d_{jm}\left(\xi_{jm},s,t\right)\;\;\;otherwise \end{array}$$

As shown in Figure 6b, the failure of a bond $\xi_{jm}$ in the peridynamic solid leads to incremental damage to the particles $x_{j}$ and $x_{m}$. As a result, the stress on the material particles could be redistributed by the force density at each loading cycle, leading to self-repeating damage to the neighboring bonds. The progressive failure of the damaged bonds leads to a crack surface 𝒫*_crack_* in the peridynamic material body. As with the original particles not in the contact region, the particle damage is described as a weighted ratio between the range of eliminated bond interactions and the general variety of provisional interactions within its horizon family. The fatigue damage of a point at an arbitrary bending stress is defined as in Figure 6b: when a bond fails in a peridynamic solid, the particles $x_{j}$ and $x_{m}$ are progressively damaged. As a result, the force density can transfer the stress to the material particles at each loading cycle, repeatedly damaging the nearby bonds. The peridynamic material body develops a fracture surface 𝒫*_crack_* as a result of the gradual collapse of the broken bonds. The damage to the particles is defined as the weighted ratio between the region of eliminated bond interactions and the overall diversity of provisional interactions within their horizon family, similar to the original particles that were not in the contact region. The fatigue damage of a point under any bending load is described as follows:(19)$$D_{x_{\left(j\right)}}=\frac{\int_{{ℋ}_{x_{\left(j\right)}}}d_{jm}\left(\xi_{jm},s,t\right)dV_{m}}{\int_{{ℋ}_{x_{\left(j\right)}}}dV_{m}}$$

Hence, between the horizon area ${ℋ}_{x_{\left(j\right)}}$ and the center point $x_{j}$, $dV_{m}$ is an incremental volume for the material particle $x_{m}$.

### 3.2. Fatigue Flexural Crack under Cyclic Bending Stress

As shown in Figure 7, during gear meshing, the driving gear transmits the load to the driven gear via the contact tooth surface. The driven gear experiences a bending moment and forms a bending stress concentration at the tooth root position (in the red circle). As the driving gear rotates continuously, the driven gear is subjected to cyclic bending stress. Over time, cracks form at the tooth root, which grow into long cracks and eventually lead to fracture. Therefore, the key to accurate tooth root stress calculation lies in the quality of the tooth contact surface transfer algorithm. In Section 2, a nonlocal peridynamic contact algorithm is proposed. Based on this contact algorithm, the transmitted load can be calculated and transferred to the driven gear.

In real peridynamic models, failure of one bond increases the stress on neighboring bonds, increasing the probability that these bonds will also break down. This leads to a progressive crack. Fractures are often arranged in two-dimensional surfaces that represent cracks. Figure 8a shows that not only bonds perpendicular to the crack surface, but also bonds with different orientations play a role in crack propagation. Crack formation and propagation occur spontaneously and autonomously, i.e., without reference to any of the flexible equations that govern these phenomena.

The direction of the crucial bonds in fracture model I is vertical, as seen in Figure 8a. Under the cyclic bending stress within the horizon, the bonds gradually fracture as the crack spreads in the x-direction. When compared to surrounding and other bonds, the strain on a certain essential bond is the greatest. The amount of critical bond strain $s_{critical}^{*}$ can be determined using the elastoplastic theory as follows:(20)$$s_{critical}^{*}\left(\delta \right)={\hat{s}}_{critical}\frac{K}{E\sqrt{\delta }}$$
where K is a dimensionless quantity that reflects the EPFM’s elastic–plastic stress intensity factor, ${\hat{s}}_{critical}$ is the so-called coefficient of strain, δ is the horizon radius and E is the elastic modulus for elastic–plastic deformation.

Figure 8b illustrates how the crucial bond and neighboring fracture produce a plastic zone $r_{p}^{*}$ close to the fatigue crack tip. The coordinate system’s value *z* is smaller than the radius of the elastic–plastic zone, which is measured in millimeters (mm). When the relationship z≫δ is satisfied, the strain distribution of the peridynamic bonds in this elastic–plastic zone approaches the strain distribution of the finite elements. According to the elastoplastic theory, the zone’s peridynamic linkages are subjected to the following strain:(21)$$s\left(z\right)=\frac{ƛ}{E\sqrt{2\pi z}}\;\;\;(0⩽z⩽r_{p}^{*})$$

Equations (20) and (21) are combined to form the following expression for a function $\hat{f}\left(\frac{z}{\delta }\right)$ that is not dependent on external cyclic loading:(22)$$\hat{f}\left(\frac{z}{\delta }\right)=\frac{1}{{\hat{s}}_{critical}\sqrt{2\pi z/\delta }}$$

Damage is often modeled in peridynamics by irreversible bond breaks. In a sense, after a bond is broken, it no longer maintains a force density. There are many types of criteria for bond base fragmentation. The simplest criterion is that the bond has uniform elongation:(23)$$s=\frac{e\langle \xi \rangle }{\left|\xi \right|}$$

A certain critical threshold $s^{*}$ is exceeded. This critical bond strain can vary depending on position, bond length, bond orientation, time, temperature or other conditions. The fatigue model described in this paper consists of a special failure criterion that does not explicitly consider the critical bond strain. Instead, each key has a history variable that characterizes the cumulative damage through multiple loading cycles.

On the basis of a one-dimensional argument, a stronger statement can be made with respect $s_{core}$ to K. The only length scale in the near-field model is the horizon δ. The dimensions of K, E, δ and $s_{core}$ can be expressed as follows:(24)[k]=Fl32,[E]=Fl2,[δ]=l,[score]=1,

Since there is only one way to obtain a dimensionless combination of the first three methods and since the material response is linear, it follows this approach:(25)$$s_{core}(\delta )={\hat{s}}_{core}\frac{K}{E\sqrt{\delta }}$$
where ${\hat{s}}_{core}$ is a dimensionless parameter independent of E, K and δ. Similarly, for type I crack tips, there must be a coordinate system and a function $\hat{f}$ that is independent of the load, which can be expressed as follows:(26)$$s({z^{\prime }}_{1},{z^{\prime }}_{2},z_{1},z_{2})=s_{core}(\delta )\hat{f}(\frac{{z^{\prime }}_{1}}{\delta },\frac{{z^{\prime }}_{2}}{\delta },\frac{z_{1}}{\delta },\frac{z_{2}}{\delta })$$

For any two points close enough to the origin, $\left(z_{1},z_{2}\right)$ and $\left({z^{\prime }}_{1},{z^{\prime }}_{2}\right)$, it has $\hat{f}\left(0,0,0,0\right)$.

If we restrict the above equation to the bonds along the axis of the type I crack, which are perpendicular and symmetrical to the crack and set, we can simplify the notation and writing:(27)$$s\left(z\right)=s_{core}\left(\delta \right)\hat{f}\left(\frac{z}{\delta }\right),\;\hat{f}\left(0\right)=1$$

The load, material properties and length scale δ are contained in the monomial $s_{core}\left(\delta \right)$, so K and E, which are not used directly in the peridynamic model, do not appear explicitly. If the distance to the crack is far enough, when z>>δ, the peridynamic bond strain field must be close to the LEFM strain field.
(28)$$s\left(z\right)∼\frac{K}{E\sqrt{2\pi z}},\;as\;z\to \infty$$

### 3.3. Peridynamic Criteria for Crack Initiation and Propagation

The mechanisms of fatigue bending cracking can be represented as two continuous phases, nucleation and propagation, as shown in Figure 9a. The strain of a bond in the nucleation phase is unrelated to the active cycle number. When a bond enters a fatigue fracture development state, its strain changes over time. When a compound goes through the fatigue fracture development phase, its strain changes. For the given material particle $x_{\left(i\right)}$, the bond $\xi_{ik}$ (as sky blue color) in the horizon of particle $x_{\left(k\right)}$ reaches the propagation phase when the damage to the particle $D_{i}(x_{i})$ meets the following conditions:(29)$$D_{i}(x_{i})\geq 0.5$$

This transition method is the same with the particle $x_{\left(j\right)}$, within the range of particle $x_{\left(m\right)}$.

## 4. Simulation Process of Tooth Root Fatigue Crack Initiation and Propagation

### 4.1. Fquivalent J-Integration between OSPD and EPFM

The load point is subjected to cyclic, uninterrupted shocks that accumulate damage in peridynamic solids. The cyclic damage of a loading cycle is simulated by the quasi-static or static solution. Therefore, the static problems of each loading cycle are defined to calculate the accumulated damage. The integral differential under bending force is the control equation for fatigue cracks.

As shown in Figure 10a, the deformation rate of three-dimensional crack J integral is defined according to the physical meaning of J integral as:(30)$$J=\underset{\Delta \theta \to 0}{\lim}\left[-\frac{dU}{dA_{0}}+\iint_{\sum \Omega }\left(T_{x}\frac{du}{dA_{0}}+T_{y}\frac{dv}{dA_{0}}+T_{z}\frac{dw}{dA_{0}}\right)dA\right]$$
where ∑Ω is a closed surface consisting of two crack front normals with angle Δθ and an arbitrary surface containing a crack tip plane, U is the deformation energy in the enclosed region surrounded by ∑Ω, $U=\underset{V}{∭}WdV$, $dA_{0}$ is the microsurface of the normal propagation of the crack front and $dA_{0}=R\Delta \theta dR$.

Based on the cylindrical coordinates, as shown in Figure 10a, Equation (30) can be expressed as follows:(31)$$\begin{array}{c} J=\underset{\Delta \theta \to 0}{\lim}\frac{1}{R\Delta \theta }[-\frac{d}{dR}\underset{V}{∭}Wrdrd\theta dz+\iint_{\Omega_{rz}}\left(\left(T_{r}\frac{du}{dR}+T_{\theta }\frac{dv}{dR}+T_{z}\frac{dw}{dR}\right)\right)rd\theta ds\\ +\iint_{\Omega_{\pm \theta }}\left(\left(T_{r}\frac{du}{dR}+T_{\theta }\frac{dv}{dR}+T_{z}\frac{dw}{dR}\right)\right)drdz] \end{array}$$

It can be observed that on the basis of the calculus mean value theorem, the $\Omega_{-\theta }$ surface, $T_{r}=-\tau_{\theta r}$, $T_{\theta }=-\sigma_{\theta }$, $T_{z}=-\tau_{\theta z}$ and on the $\Omega_{+\theta }$ surface, $T_{r}=\tau_{\theta r}+\frac{\partial \tau_{\theta r}}{\partial \theta }d\theta$, $T_{\theta }=\sigma_{\theta }+\frac{\partial \sigma_{\theta }}{\partial \theta }d\theta$, $T_{z}=\tau_{\theta z}+\frac{\partial \tau_{\theta z}}{\partial \theta }d\theta$. Then, Equation (31) can be simplified as follows:(32)$$\begin{array}{r} J=\frac{1}{R}[-\frac{d}{dR}\iint_{\Omega_{\theta }}Wrdr+\oint_{C_{\theta }}\left(T_{r}\frac{du}{dR}+T_{\theta }\frac{dv}{dR}+T_{z}\frac{dw}{dR}\right)rds\\ +\iint_{\Omega_{\theta }}\frac{\partial }{\partial \theta }\left(\tau_{\theta r}\frac{du}{dR}+\sigma_{\theta }\frac{dv}{dR}+\tau_{\theta r}\frac{dw}{dR}\right)drdz] \end{array}$$

According to Figure 10b, an analogous stress intensity factor along the crack front is defined as follows to establish a correlation between the flexural fatigue crack tip in OSPD and EPFM:(33)$$\begin{array}{r} K^{equiv}=\frac{1}{R}[\int_{\Gamma_{\theta }}Wrdz-\left(T_{r}\frac{\partial u}{\partial r}+T_{\theta }\frac{\partial v}{\partial r}+T_{z}\frac{\partial w}{\partial r}\right)rds\\ -\iint_{\Omega_{\theta }}\frac{\partial u}{\partial \theta }\left(\tau_{\theta r}\frac{\partial u}{\partial r}+\sigma_{\theta }\frac{\partial v}{\partial r}+\tau_{\theta z}\frac{\partial w}{\partial r}\right)drdz] \end{array}$$
where $T_{\theta }$ is an integral path around the crack tip in a counterclockwise orientation on the normal plane of the crack front from any location on the lower surface of crack to any position on the upper surface, $\Omega_{\theta }$ is a region surrounded by the crack tip boundary and $\Gamma_{\theta }$ on the normal plane of the crack front, W is the deformation energy density at any particle on the path $\Gamma_{\theta }$ and this deformation energy density can be calculated as follows:(34)$$W=\int \sigma_{r}d\varepsilon_{r}+\sigma_{\theta }d\varepsilon_{\theta }+\sigma_{z}d\varepsilon_{z}+\tau_{r\theta }d\gamma_{r\theta }+\tau_{\theta z}d\gamma_{\theta z}+\tau_{zr}d\gamma_{zr}$$

### 4.2. Numerical Method for Static Solution

When certain bonds expire for the duration of a charge cycle, a new static solution is derived for the current cycle. For a given particle, the equation for the motion manipulation is a governing differential equation with notional inertia terms. As for all particles in a structure, a set of equations is expressed as follows:(35)$$D\ddot{U}(X,t)+\zeta D\dot{U}(X,t)=F(U,U^{\prime },X,X^{\prime })$$
where X is the initial position vector, U is the first motion vector, ζ is the damping coefficient and D is the virtual diagonal density matrix. X and U are applied as follows to material particles in the peridynamic solid:(36)$$\left\{ \begin{array}{l} X^{T}=\left\{x_{1},x_{2},\cdots ,x_{N}\right\}\\ U^{T}=\left\{u(x_{1},t),u(x_{2},t),\cdots ,u(x_{N},t)\right\} \end{array}\right.$$
where the configuration body’s total number of material points is given by the integer N. Last but not least, the vector F is made up of body forces and PD contact and its k^th^ component can be expressed as follows:(37)$$\{\begin{array}{l} V^{n+1/2}=\frac{\left(\left(2-\zeta^{n}\Delta t\right)V^{n-1/2}+2\Delta tD^{-1}F^{n}\right)}{2+\zeta^{n}\Delta t}\\ U^{n+1}=U^{n}+V^{n+1/2}\Delta t \end{array}$$
where n represents the total number of iterations. The density matrix’s diagonal, D, elements are implemented as follows:
(38)$$\gamma_{kk}⩾\frac{1}{4}\Delta t^{2}\sum_{j}\left|J_{kj}\right|$$

If the stiffness matrix $J_{kj}$ of the peridynamic material is solid under the additional tiny placement hypothesis, it is then written as follows:(39)$$\sum_{j}\left|J_{kj}\right|=\sum_{j=1}^{M}\frac{\left|\xi_{\left(k\right)\left(j\right)}\cdot e\right|}{\left|\xi_{\left(k\right)\left(j\right)}\right|}\frac{4\delta }{\left|\xi_{\left(k\right)\left(j\right)}\right|}\left\{\frac{ad^{2}\delta }{\left|\xi_{\left(k\right)\left(j\right)}\right|}(v_{ck}V_{k}+v_{cj}V_{j})+b\right\}$$
where the governing constants a, b and d are present and e is a unit vector pointing in the direction of the nondiagonal. Summarizing the results will yield the stiffness matrix’s constituent parts. Equations (37) and (35) can be used to express the damping coefficient as:(40)ζn=2((Un)TKn1Un)/((Un)TUn)
where Kn1 is the diagonal “local” stiffness matrix, which is denoted as follows:(41)$${}^{1}K_{ii}^{n}=-\left(F_{i}^{n}/\lambda_{ii}-F_{i}^{n-1}/\lambda_{ii}\right)/\left(\Delta t{\dot{u}}_{i}^{n-1/2}\right)$$

### 4.3. Mesh Sensitivity in Peridynamic Static Solution

To study δ convergence, the horizon must contract while the distance between particles remains constant. A particle cannot be connected to other particles in a discretized body if δ is smaller than the distance between them. For this study, the largest particle distances were considered up to the five largest particle distances; larger sizes were not considered because of the increasing computational cost. The authors expanded all horizons by one percent because it is advisable to do so to avoid omitting particles from a family due to a floating-point error.

Cases are denoted as lmn for simplicity, where l is the number of samples, m is the number of particles in the sample’s thickness (1 to 5) and n is the size of the horizon in grids (1 to 5). Table 1 provides information on the particle and horizon sizes.

As shown in Figure 11, according to the δ convergence study, results are most accurate when the horizon size is three-times the grid size and accuracy declines as the horizon size is increased more. The findings of five compression cases with horizon sizes of $\sqrt{2}h$ were comparable to or superior to those of cases with larger horizon sizes. This would suggest that using horizon sizes other than those equal to the grid size times an integer can produce results that are more accurate, but additional research is required.

Despite varying grid sizes, the horizon to grid size ratio of three produces the best accurate findings in the samples, according to the (δh) convergence study. At greater particle spacings, horizon size $\sqrt{2}h$ for compression is most accurate, but as particle spacings become smaller, a horizon size of three grids produces more accurate results.

### 4.4. The Process to Simulate Tooth Root Fatigue Crack Initiation and Propagation

The OSPD model associates material particles with precise volumes in a region that is uniformly discretized. Solving the PD equations of motion provides confirmation of the structure’s response under static severe loading. The beginning and spread of fatigue cracks can be depicted in the common state-based theoretical framework PD, as illustrated in Figure 12, in a manner similar to the classical process in the EPFM framework.

## 5. Model Verification Based on Experimental Results

### 5.1. Fatigue Test Equipment

As seen in Figure 13, the wind turbine transforms wind energy through the impeller into rotational kinetic energy, which is subsequently transferred to the generator through the transmission system to produce electricity. Due to the low speed of the wind-driven runner (typically 5 to 22 rpm), the speed of the gearbox must be increased to a high speed suitable for operating the generator. The gearbox system, the heart of the wind turbine, essentially consists of the runner, the spindle, the speed-increasing gearbox, the generator and the control system.

To confirm the damage of the wind turbine gearbox under wind action, a full-scale prototype test system was built in the laboratory, as shown in Figure 14. The test system mainly consists of three parts: (a) Fan simulation part. The power consumption with time-varying low speed and high torque is achieved by driving the reduction gearbox with a frequency conversion motor. (b) Speed-increasing gearbox. The gearbox used in the experimental platform has the same structural characteristics as the research object of the project, a scale model with a power of 15 kW. The load motor of the simulated generator uses the torque control method, so the frequency converter can be used to simulate the actual operation process of the wind power gearbox under operating conditions. The scaled prototype test system uses a typical “back-to-back” structure, which is also widely used in various industrial gearbox test benches.

### 5.2. Sample Dimension and Input Parameters

The relative percentage contents of elements in the special alloy steel material are shown in Table 2. The main mechanical property parameters of the special alloy steel material at room temperature are shown in Table 3.

### 5.3. Results Comparison and Analysis

As shown in Figure 15, the use of the macroscopic crack remark method and information recording generation, it can be seen that the driven gear has a tooth root crack. The cycle numbers of the tooth root crack are recorded in Table 4.

The average fatigue cycle number for the occurrence of the tooth root fracture is 276,154, as indicated in Table 4 for cycle numbers throughout commencement. Thus, 268,136 is the average number of fatigue cycles for all propagation. The total lifespan of the equipment fatigue tooth crack comes to 544,290 when these lifestyle periods are added together.

We can observe that the angle between the fatigue crack and the base plane is approximately 60.5° when comparing Figure 15’s illustration of the fatigue crack’s path to Figure 16. The average test result and this simulation result have a good correlation, as seen in Table 4.

As shown in Figure 17, the value of the equivalent stress intensity factor $K^{equiv}$ becomes larger as the number of iterations increases. The results show that as the initial crack length increases, the stress intensity factor also increases and the stress intensity factor increases proportionally to the load. Under the same initial crack length and load, the stress intensity factor $K_{I}$ is much larger than $K_{\mathrm{II}}$ and $K_{\mathrm{III}}$; that is, the open crack is the main reason for the fracture failure of wind power gear teeth under bending stress. According to Figure 17, when the number of iterations rises, the value of the equivalent stress intensity factor $K^{equiv}$ increases. The findings demonstrate that the stress intensity factor increases proportionally to the load and increases along with the first fracture length. The open crack is the main cause of the fracture failure of wind power gear teeth under bending stress because, for the same starting crack length and load, the stress intensity factor $K_{I}$ is significantly bigger than $K_{\mathrm{II}}$ and $K_{\mathrm{III}}$.

## 6. Conclusions

(1)A novel OSPD fatigue model for the initiation and propagation of fatigue cracks at the tooth root was derived to evaluate the service life of the driven gear under bending fatigue loading. The fatigue crack at the tooth root germinates and propagates independently with this constitutive fatigue model.(2)The application of the entire fatigue crack propagation in the tooth root to the suggested damage model of ordinary state-based peridynamics is possible because the model has no size limitations. In light of this, the OSPD fatigue model has effectively taken into account cross-scale issues that may arise during the lifetime of fatigue fractures in tooth roots.(3)According to the time record, the tooth root crack germinates and grows into larger fissures. The proposed version’s numerical calculation results and the outcomes of the experiment show good agreement. According to our comparison, it is more effective and accurate than standard fatigue models at reproducing the tooth root fracture features as well as the spatial displacement of individual positions.(4)Without extra guidelines for manual crack propagation, the natural production and propagation of fatigue cracks in the tooth root is confirmed. A quantitative analysis of fatigue damage is performed. The evaluation of three-dimensional nucleation of fatigue cracks in the tooth root to predict fatigue life is confirmed based on the OSPD version.

## Figures and Tables

**Figure 1 materials-15-07762-f001:**
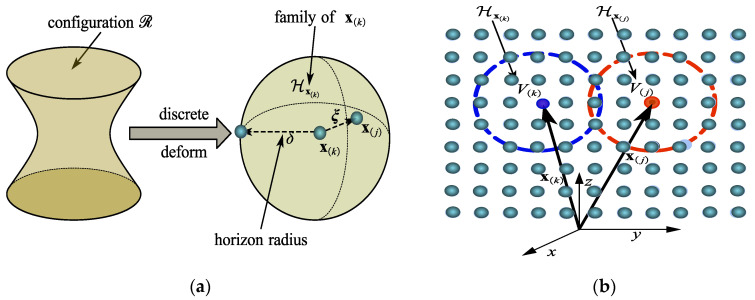
Kinematics of state-based PD material particles. (**a**) Peridynamic body and particle horizon; (**b**) motion of material particle within its horizon.

**Figure 2 materials-15-07762-f002:**
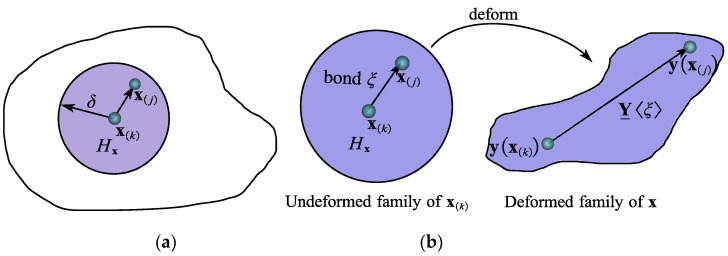
The force state of state-based peridynamic theory. (**a**) Example peridynamic domain; (**b**) deformation state $\underline{Y}\langle \xi \rangle$ acting on bond ξ.

**Figure 3 materials-15-07762-f003:**
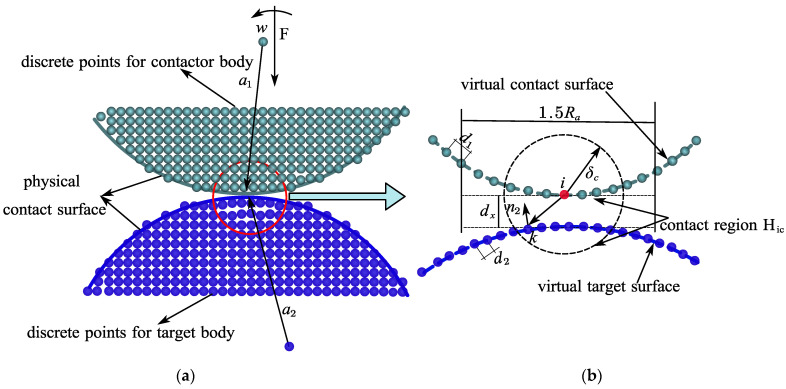
(**a**) Physical contact faces between two surfaces. (**b**) Peridynamic virtual contact surface and target surface.

**Figure 4 materials-15-07762-f004:**
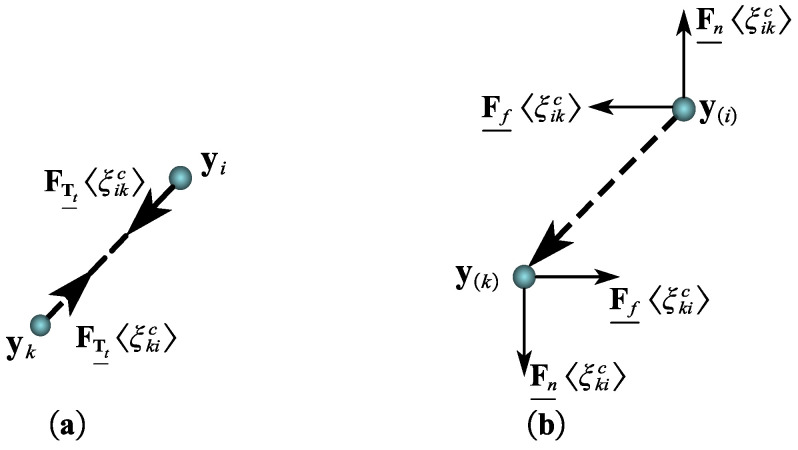
The formation of peridynamic contact pair in two conditions. (**a**) Sticking. (**b**) Sliding.

**Figure 5 materials-15-07762-f005:**
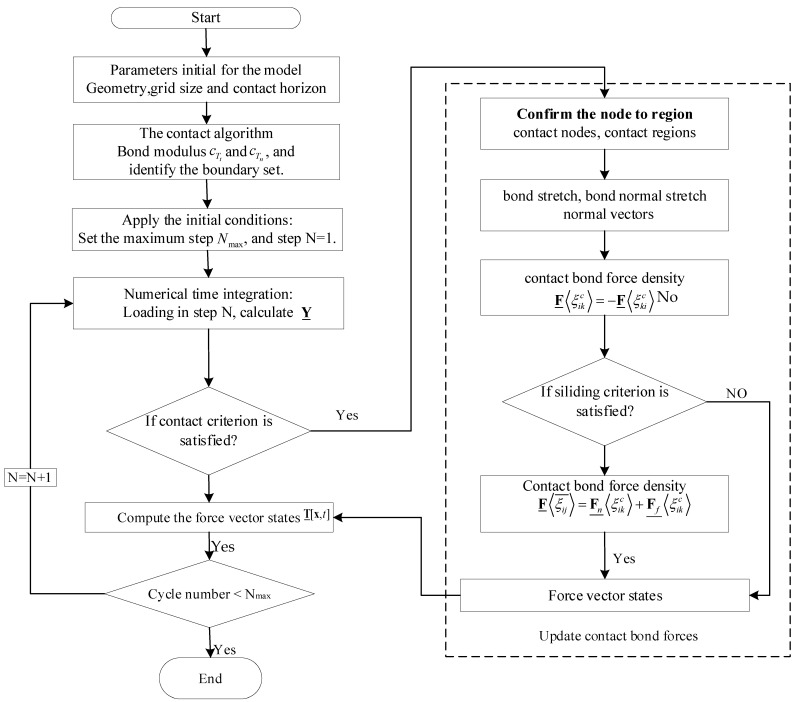
The flowchart of the nonlocal peridynamic contact algorithm.

**Figure 6 materials-15-07762-f006:**
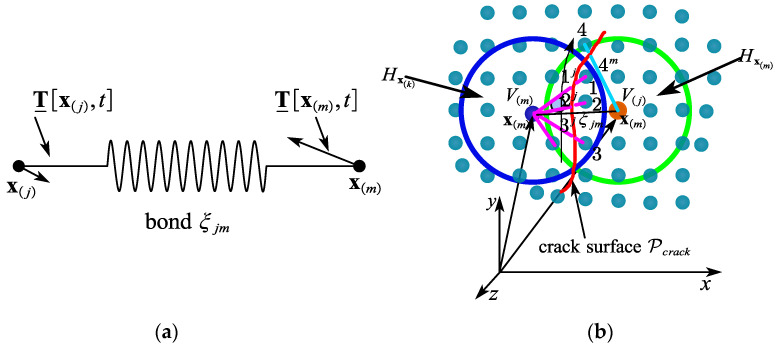
The progressive failure leading to crack surface. (**a**) The force on peridynamic bond. (**b**) Progressive failure of peridynamic bond.

**Figure 7 materials-15-07762-f007:**
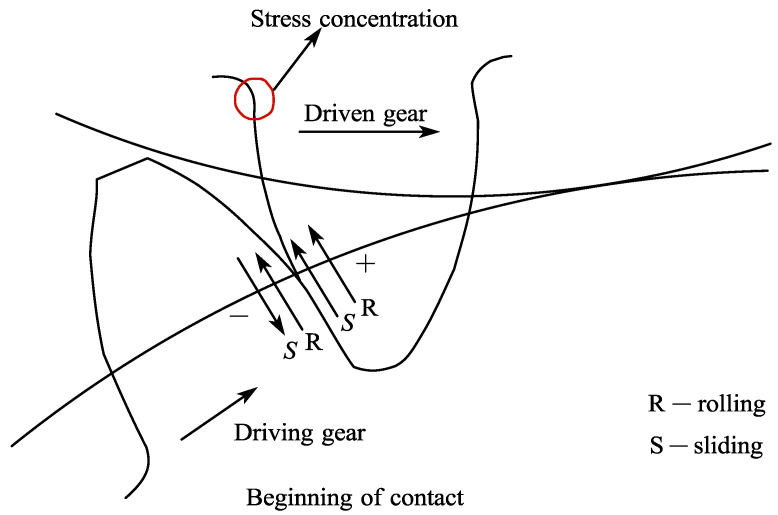
Contact surface between driven and driving gear.

**Figure 8 materials-15-07762-f008:**
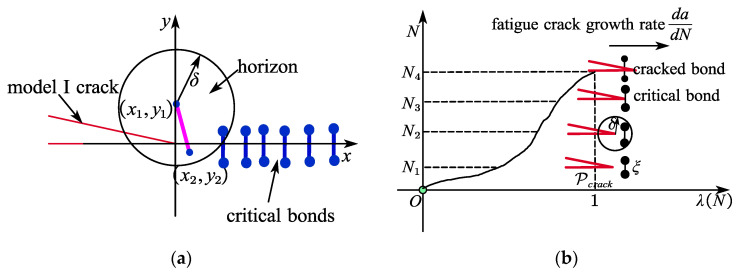
The peridynamic fatigue crack under bending loads. (**a**) The critical bond in the model I crack. (**b**) The bond breaks in a progressive way.

**Figure 9 materials-15-07762-f009:**
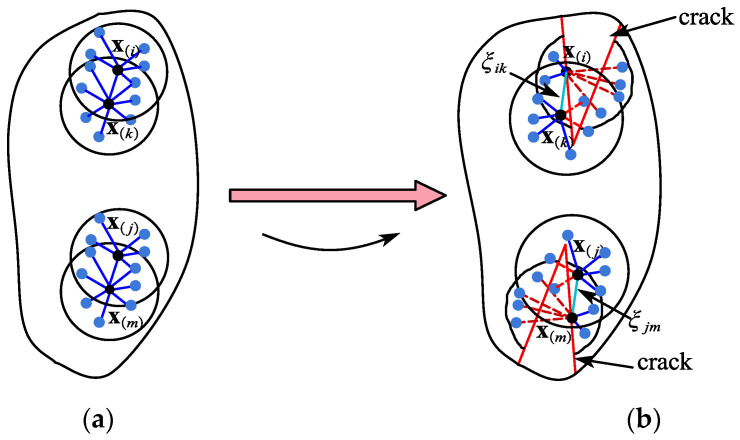
Transition from tooth root fatigue crack nucleation to propagation. (**a**) Nucleation of fatigue crack. (**b**) Propagation of fatigue crack.

**Figure 10 materials-15-07762-f010:**
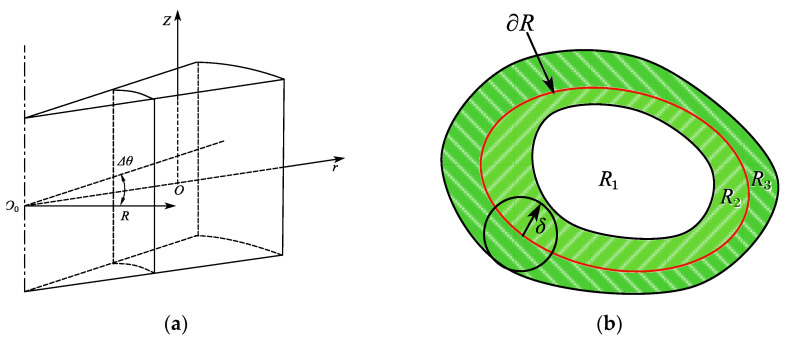
Equivalent *J*-integration based on peridynamic theory. (**a**) Three-dimensional crack *J* integral region. (**b**) Equivalent stress factor.

**Figure 11 materials-15-07762-f011:**
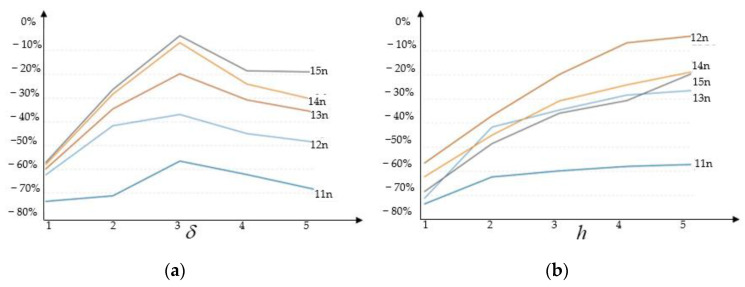
Mesh sensitivity under different horizons and grid sizes. (**a**) Percent error vs. horizon size (δ convergence). (**b**) Percent error vs. particle spacing (δh convergence) graphs.

**Figure 12 materials-15-07762-f012:**
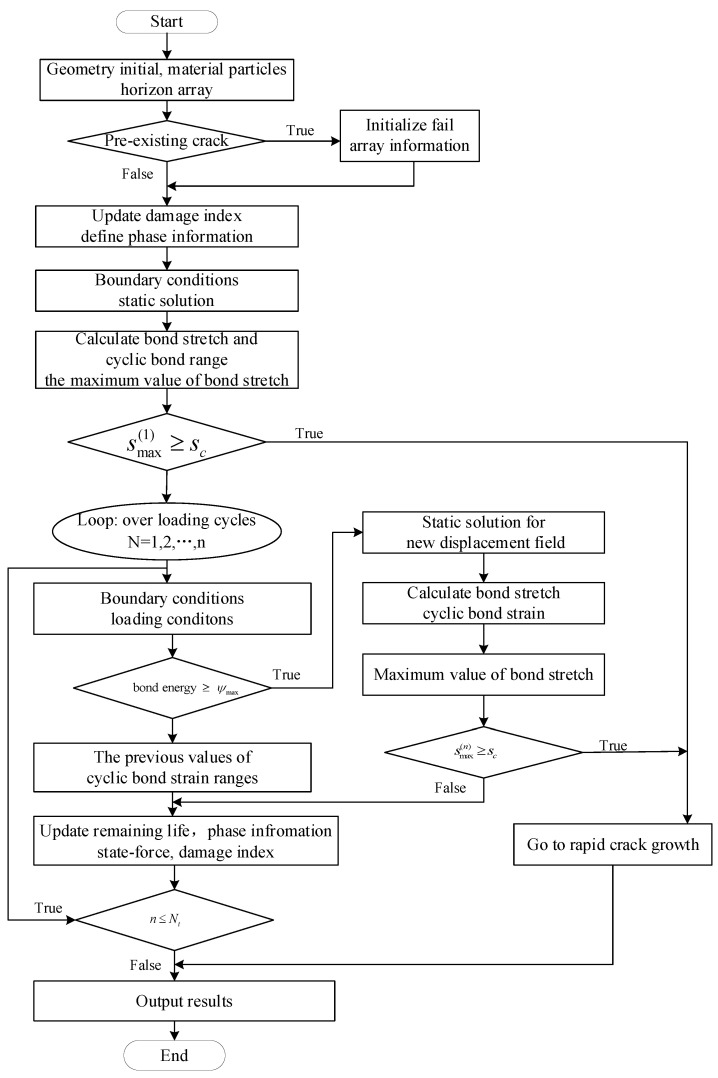
Follow chart of bending crack simulation.

**Figure 13 materials-15-07762-f013:**
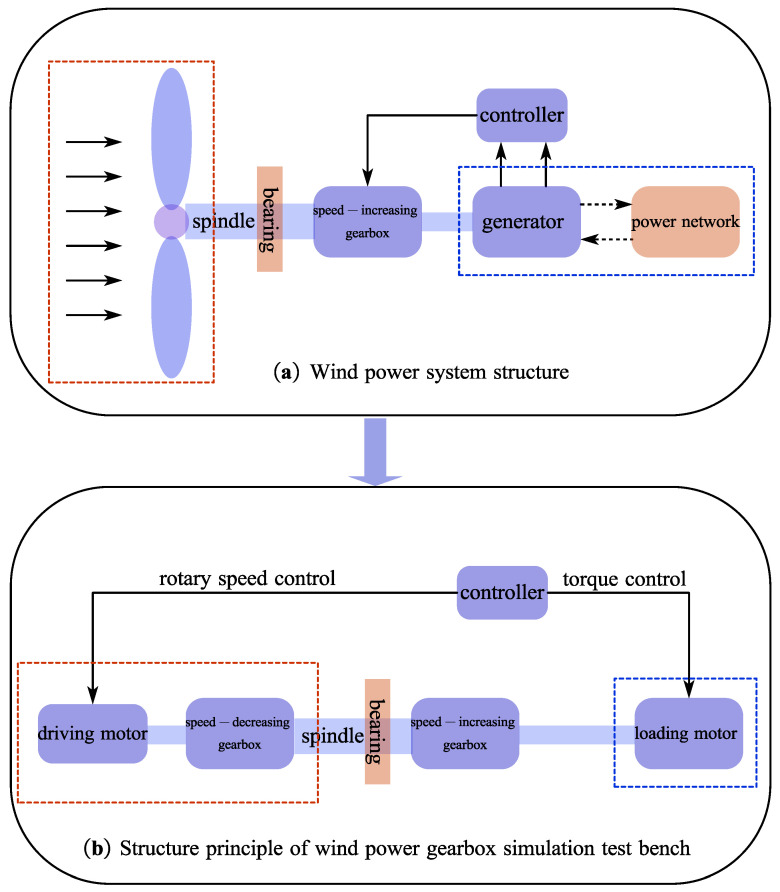
Gear fatigue test platform schematic diagram.

**Figure 14 materials-15-07762-f014:**
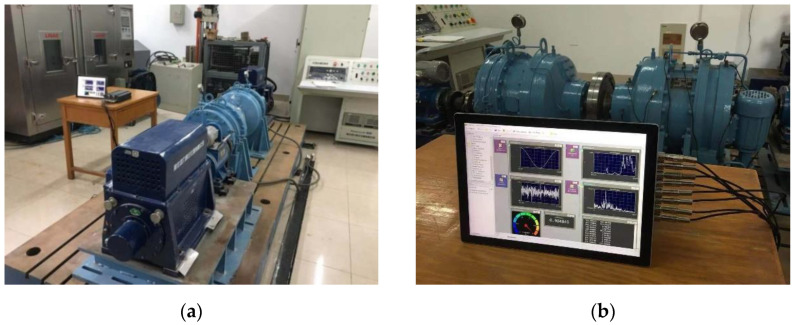
Bending fatigue test center. (**a**) Gearbox test bench. (**b**) Data acquisition and display.

**Figure 15 materials-15-07762-f015:**
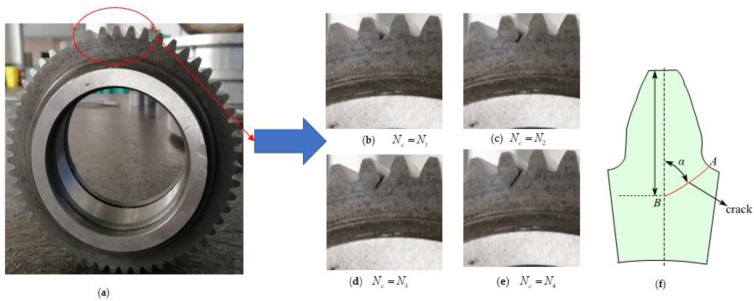
Marco tooth root fatigue crack of the test specimen. (**a**) driven gear sample; (**b**) crack propagates under cyclic index *N*_c_ = *N*_1_; (**c**) crack propagates under cyclic index *N*_c_ = *N*_2_; (**d**) crack propagates under cyclic index *N*_c_ = *N*_3_; (**e**) crack propagates under cyclic index *N*_c_ = *N*_4_; (**f**) crack propagates along the red lines.

**Figure 16 materials-15-07762-f016:**
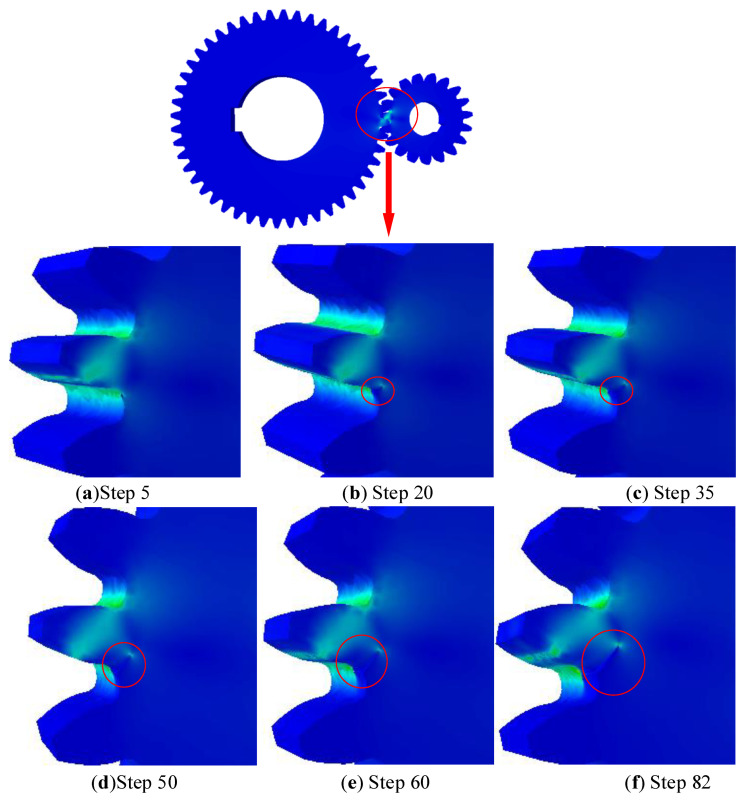
Simulation results of bending crack based on OSPD fatigue model. (**a**) Crack propagates under iterative step = 5; (**b**) crack propagates under iterative step = 20; (**c**) crack propagates under iterative step = 35; (**d**) crack propagates under iterative step = 50; (**e**) crack propagates under iterative step = 60; (**f**) crack propagates under iterative step = 82; The red circles in the figures indicate the crack length is gradually increasing.

**Figure 17 materials-15-07762-f017:**
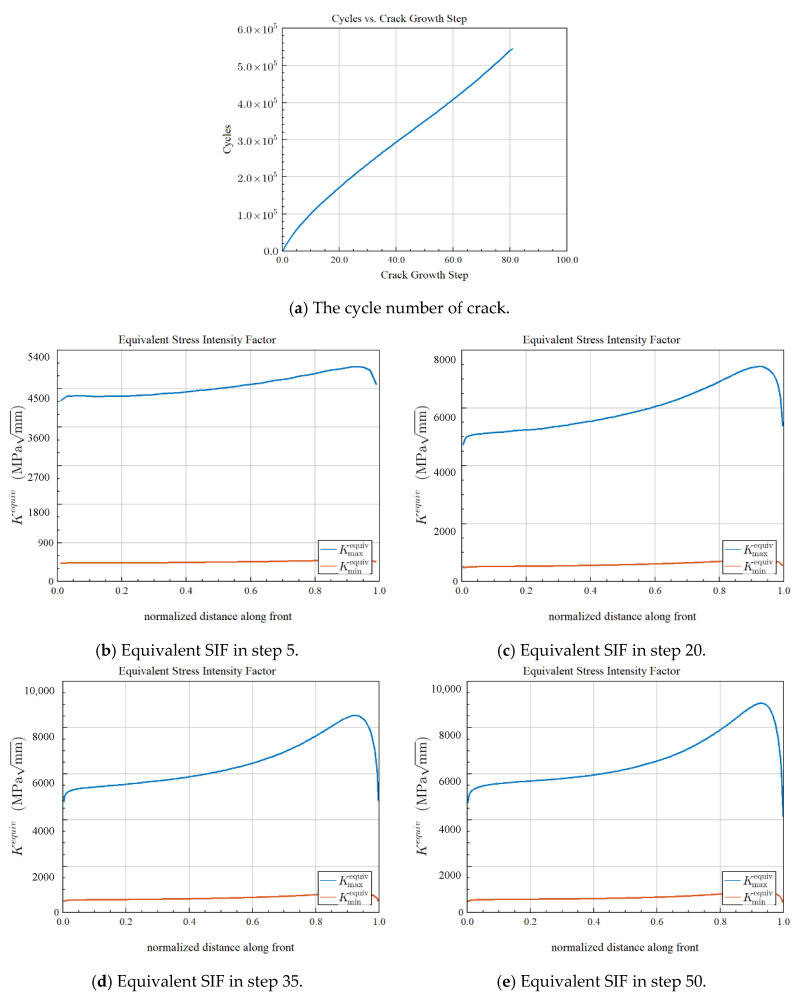
In each specified iteration, the equivalent intensity factors along crack front. (**a**) Cycle number vs. crack growth step; (**b**) $K^{equiv}$ along crack propagation front with iterative step = 5; (**c**) $K^{equiv}$ along crack propagation front with iterative step = 20; (**d**) $K^{equiv}$ along crack propagation front with iterative step = 35; (**e**) $K^{equiv}$ along crack propagation front with iterative step = 50; (**f**) $K^{equiv}$ along crack propagation front with iterative step = 60; (**g**) $K^{equiv}$ along crack propagation front with iterative step = 82.

**Table 1 materials-15-07762-t001:** Particle (h) and horizon (δ) sizes for all instances taken into consideration. Each specimen’s mesh density and five various horizon diameters are taken into account.

Cases	$h_{x}\;(m)$	$h_{y}\;(m)$	$h_{z}\;(m)$
l1n(1…5) × 1.01 × $h_{y}$	0.002068	0.002400	0.001958
l2n(1…5) × 1.01 × $h_{y}$	0.001048	0.00105	0.001020
l3n(1…5) × 1.01 × $h_{y}$	0.000669	0.000700	0.000684
l4n(1…5) × 1.01 × $h_{y}$	0.000494	0.000516	0.000534
l5n(1…5) × 1.01 × $h_{y}$	0.000389	0.00042	0.000434

**Table 2 materials-15-07762-t002:** Related parameters of standard spur gear.

Gear	Tooth Number	Module/mm	Pressue Angle	Face Width/mm
Driving gear	19	5	20	40
Driven gear	48	5	20	40

**Table 3 materials-15-07762-t003:** Main mechanical property parameters of special alloy steel material at room temperature.

Name	E/GPa	σ_0.2_/MPa	σ_b_/MPa	Fatigue Strength/MPa	Density g/cm^3^	Poisson’ Ratio	Brinell Hardness/HB	Shear Modulus/GPa	Shear Strength/MPa
18CrNiMo7–6	210	580	795	320	3.0	0.3	229	80	330

**Table 4 materials-15-07762-t004:** Cycle numbers and angles recorded for tooth root fatigue cracks.

Samples	Initiation	Propagation	Angle
18CrNiMo7-6-1	298,910	286,410	62.5
18CrNiMo7-6-2	285,334	300,234	65.5
18CrNiMo7-6-3	300,368	265,036	59.3
18CrNiMo7-6-4	290,658	29,058	61.8
18CrNiMo7-6-5	300,648	200,489	58.8
18CrNiMo7-6-6	241,006	268,586	62.6

## Data Availability

The data sets generated and/or analyzed during the current study are available from the corresponding author on reasonable request.

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
