# Peer review of "A State-Based Peridynamic Flexural Fatigue Model for Contact and Bending Conditions"

_materials, 2022, doi:10.3390/ma15217762_

Round 1

Reviewer 1 Report

The authors have developed a state-based peridynamic approach for fatigue analysis applied to gear contacts. At first sight the article looks fine and balanced, however, after closer inspection there are many small things to be corrected or clarified.

Below are my observations:

  • The introduction is long and well written except that the citations in many cases do not match, e.g. like on line 167 – Silling et al. [60], there are also many others for which the author and number do not match.

  • Equation (1), why? It is not used in the manuscript at all!

  • In equation (2) the notation of the angle brackets is not defined.

  • Line 262 the authors write: “If Equation (7) is...”, it should be: “If inequalities (7) are ...”

  • Line 264 and equation (8), the definition of the contact region Hic is unclear.

  • Line 275 the authors write “fig 4a …. the contact bond is used as thick dashed line ...”. I do not see any dashed lines in figure 4.

  • After equation (10) on line 282 the authors claim that T_t and T_n are state forces, however on line 293 the authors claim that they are strains of the bond and on line (296) sticking and sliding micromodulus?

  • The coefficients c_t, c_n are not defined, they are probably the sticking and sliding modules.

  • How the values for these mocromodules are obtained?

  • What is the affect function omega?

  • Equation (13) is ununderstandable.

  • In figure 5 the conditional boxes lack the Yes/No labels.

  • I do not understand the sentences on line 323-325.

  • On line 342 it is mentioned “bending stress” also in some other places. Why to emphasize “bending”, is there some difference to “normal stress” which contains stress from both bending moment and axial forces?

  • In figure 7 the symbols R-rolling, S-sliding? Should they appear also in the figure somewhere?

  • Line 365 “flexible equations”?

  • Figure 8 caption: “flexural fatigue crack”?

  • I do not understand the development on lines 377-418.

  • On lines 425-427 the authors write “the bond reaches the propagation stage when the damage of the particle damage exceeds 0.5? Before it was mentioned that it is autonomous. Which is true and how this particular value is chosen.

  • On line 459 the boundary part is incorrectly mentioned as T_theta. It should be Gamma_theta.

  • On line 472 the authors call the damping coefficient zeta as bulk drag coefficient, how it is determined?

  • In section 5.4 there are many citation errors where reference source not found.

As there are so many mistakes, unclear parts, unfortunately, I do not recommend publication of the manuscript.

Reviewer 2 Report

The paper investigated fatigue life under bending. Also, they modeled contact. The text is clear and easy to read. The paper well written. It can be published after the revise.

1- Please add a notation list.

2- Please comment on the fatigue life prediction. 

3- It cannot be seen the crack. Please add a photo and show the crack.

4- Please emphasis on the novelty of this paper in Introduction Section.

5- The contact modeling is an important problem in the works on the fatigue. Please more explain on the details of the contact modeling. For example, which surface is selected as target or contact plane?

6- Analyses of sensitive mesh size should be conducted and added in the paper.

Reviewer 3 Report

Authors have studied State-based Peridynamic Flexural Fatigue Model for Contact and Bending Conditions. The study will add value to the field. 

There are some formatting issues such as "Error! Reference source not found." in the results and comparison section. 

It is recommended to introduce studies for a comparative report on results available in literature and the results obtained in the study. 

The validation of the modelling through experiments should be explained thoroughly (if possible). 

Authors are recommended to introduce following state of the arts in the manuscript to strengthen discussion

https://doi.org/10.1016/j.engfracmech.2022.108362

https://doi.org/10.1016/j.compscitech.2022.109303

https://doi.org/10.3390/math9050507

  • 10.1109/ITHERM.2010.5501273
  • Good luck

Round 2

Reviewer 2 Report

The paper suggests for publication.